# Transformers Use Causal World Models in Maze-Solving Tasks

## Abstract

Recent studies in interpretability have explored the inner workings of transformer models trained on tasks across various domains, often discovering that these networks naturally develop surprisingly structured representations. When such representations comprehensively reflect the task domain's structure, they are commonly referred to as "World Models" (WMs). In this work, we discover such WMs in transformers trained on maze tasks. In particular, by employing Sparse Autoencoders (SAEs) and analysing attention patterns, we examine the construction of WMs and demonstrate consistency between the circuit analysis and the SAE feature-based analysis. We intervene upon the isolated features to confirm their causal role and, in doing so, find asymmetries between certain types of interventions. Surprisingly, we find that models are able to reason with respect to a greater number of active features than they see during training, even if attempting to specify these in the input token sequence would lead the model to fail. Futhermore, we observe that varying positional encodings can alter how WMs are encoded in a model's residual stream. By analyzing the causal role of these WMs in a toy domain we hope to make progress toward an understanding of emergent structure in the representations acquired by Transformers, leading to the development of more interpretable and controllable AI systems.

## 1 Introduction

The study of world models (WMs) in AI systems has gained significant traction of late, yet much interpretability research focuses on large language models trained on diverse, complex datasets (Belrose et al., 2023; Lieberum et al., 2023; Olsson et al., 2022). In an attempt to seek a more comprehensive understanding, our work examines WMs acquired by Transformers (Vaswani, 2017) in a controlled, synthetic environment. In particular, we use Maze-solving tasks (Subsection 2.1) as an ideal testbed for understanding learned WMs due to their human-understandable structure, controllable complexity, and relevance to spatial reasoning. Using this constrained domain, we can rigorously analyze how transformers trained (Subsection 2.2) to solve mazes construct and utilize internal representations of their environment.

Our methodology leverages Sparse Autoencoders (SAEs) (Bricken et al., 2023) to overcome the limitations of linear probes in detecting WM features. While linear probes can and have been used to identify latent directions associated with features defined in an imposed ontology, SAEs are found to discover features actually used by our models to make decisions (Section 3). By manipulating specific features identified by the SAEs and observing the impact on our models' maze-solving behavior, we provide strong evidence that these features are causally involved in the model's decision-making process (Section 4). This stands in contrast to prior work analyzing the formation of WMs in maze settings where no causal features were able to be isolated (Ivanitskiy et al., 2024).

Our findings provide important considerations for AI interpretability and alignment. By investigating how transformers form causal WMs even in relatively simple tasks, we hope to provide new avenues for understanding representations and potentially steering behavior in more complex AI systems. This work lays the groundwork for future research into how we might intervene on WMs to better align transformer-based AI systems to desired constraints.

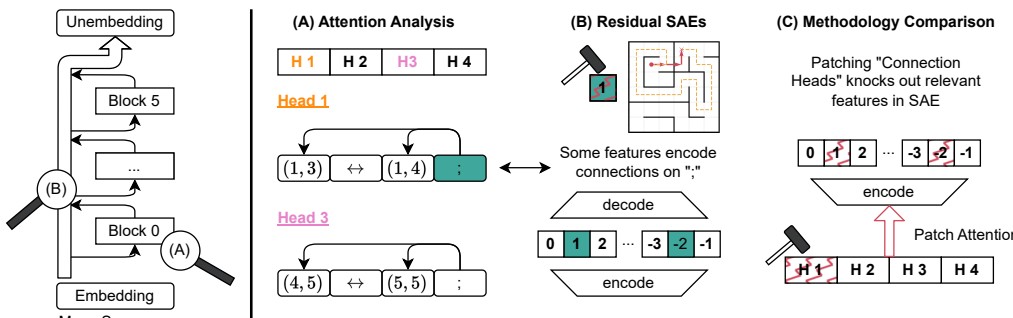

Figure 1: Overview of our methodology for discovering and validating world models in transformer-based maze solvers. **(A)** We analyze attention patterns in early layers, finding heads that consolidate maze connectivity information at semicolon tokens. **(B)** We train sparse autoencoders on the residual stream immediately following the first block, identifying interpretable features that encode maze connectivity. **(C)** We demonstrate the causal role of the world models in our transformers comparing the features extracted through both methods and validating them through causal interventions.

We outline our methodology in Figure 1. In short, we begin in Subsection 3.1, identifying attention heads that appear to construct world model features by examining their attention patterns across maze coordinate tokens (A). We validate these findings in Subsection 3.3 by training sparse autoencoders on the residual stream and demonstrating that the extracted features match those identified through attention analysis (B). Lastly, in Subsection 3.3 we establish the causal nature of these representations through targeted interventions, showing that perturbing specific features produces predictable changes in the model's maze-solving behavior (C).

**Contributions**

- **Empirical Findings:** We show that transformers form WMs when solving mazes and that these WMs are causal: they can be intervened upon in the latent space of SAEs. Surprisingly, we find that interventions that activate features are more effective than those that remove them, suggesting an asymmetry in how transformers utilize WM features. In performing these interventions we also uncovered our models' abilities to reason in the presence of a out-of-distribution number of activated features than would naturally arise for a given token sequence length.

- **Methodological Insights:** By effectively utilizing decision trees to isolate WMs features in SAEs, we demonstrate that transformers utilizing different encodings schemes may use varyingly compositional codes to represent their WMs. More generally, our analyses suggest that SAEs are generally better suited than linear probes to isolate WMs, even in the absence of feature splitting.

## 2 PRELIMINARIES

### 2.1 ENVIRONMENT

Though it remains a matter of debate whether Large Language Models (LLMs) construct structured internal models of the real world, we can begin to understand the representations acquired by such models by focusing on "toy" tasks with clear spatial or temporal structure (Brinkmann et al., 2024; Momennejad et al., 2024; Jenner et al., 2024; McGrath et al., 2022). Previous works along these lines (Li et al., 2022; Ivanitskiy et al., 2024; Nanda, 2023; Karvonen, 2024; He et al., 2024) have found a variety of both correlational and causal evidence for internal models of the environment within trained transformers. In this work, we utilize `maze-dataset` (Ivanitskiy et al., 2023), a package providing maze-solving tasks as well as ways of turning these tasks into text representations. In particular, we use a dataset of mazes consisting of up to $7 \times 7$ grids, generated via constrained randomized depth first search (see Subsection 2.2) (which produces mazes that are acyclic and thus have a unique solution).

To train autoregressive transformers to solve such mazes, we employed a tokenization scheme provided by `maze-dataset`, shown in Figure 2. This scheme is designed to present the maze structure, start and end points, and solution path in a format amenable to transformer processing whilst remaining straight-forward to analyse with standard tools from the mechanistic interpretability literature - primarily due to the existence of a unique token for every position in the maze (aka "lattice").

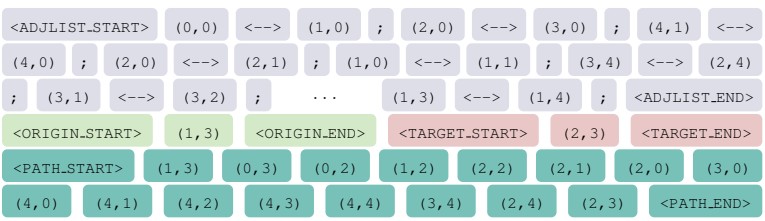
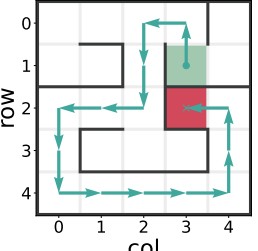

(a) An example of a tokenized maze. **1:** The adjacency list describes the connectivity of the maze, with the semicolon token `;` delimiting consecutive connections. The order of connections is randomized, ellipses represent omitted connection pairs. **2,3:** The origin and target specify where the path should begin and end, respectively. **4:** The path itself a sequence of coordinate tokens representing the shortest path from the origin to the target. For a "rollout," we provide everything up to (and including) the `<PATH_START>` token and autoregressively sample with `argmax` until a `<PATH_END>` token is produced.

(b) Visual representation of the same maze as in the tokenized representation on the left. The origin is indicated in green, the target in red, and the path in blue.

Figure 2: Tokenization scheme and visualization of a shortest-path maze task generated using Ivanitskiy et al. (2023).

## 2.2 MAZE SOLVING TRANSFORMERS

Utilizing the tokenized representations of mazes provide by the `maze-dataset` library, a suite of transformer models implemented using TransformerLens ((Nanda & Bloom, 2022)) were trained to predict solution paths in acylic mazes. We performed extensive hyperparameter sweeps (Figure 10) over several variants of the transformer architecture, yielding models with stronger generalization performance than those found by prior work Ivanitskiy et al. (2024).

To allow the testing of generalization to large maze size, the models were trained on $5 \times 5$ fully-connected and $6 \times 6$ sparsely connected mazes, embedded in a $7 \times 7$ lattice. This ensured that all coordinate tokens in the $7 \times 7$ vocabulary had been seen during training time, such that generalization to $7 \times 7$ mazes was conceivable but out-of-distribution during inference. For our experiments, we investigated the two best performing models for each positional embedding (Su et al., 2024) scheme, as shown in Table 1. Note that whilst these models had different numbers of heads, their parameter counts varied only slightly - on account of Stan's use of learned positional embeddings.

## 3 DISCOVERING WORLD MODELS

Broadly speaking there are two ways to go about trying to identify internal world models: 1) Assuming the form of the world model and inspecting the transformer with e.g. supervised probes to see if this world model is present ((Nanda, 2023) SELF CITE Workshop proceeding), or 2) Exploring the model internals and investigating any structure which may be present in the representations to see if something akin to a world model exists. In our work we take both approaches.

| Model Nickname | Positional Embeddings | $d_{\text{model}}$ | $n_{\text{layers}}$ | $n_{\text{heads}}$ | Num. Params | Maze Solving Accuracy |
|---|---|---|---|---|---|---|
| Stan | **stan**dard learned | 512 | 6 | 8 | 19,225,660 | 96.6% |
| Terry | ro**tary** | 512 | 6 | 4 | 18,963,516 | 94.3% |

Table 1: Models chosen for mechanistic investigation (most performant in the sweep, given their respective position embedding schemes). The number of parameters varies as Stan learns position encodings ($W_{pos} \in \mathbb{R}^{512 \times 512}$)

First, in Subsection 3.1 we investigate attention heads in the earliest layer of our models and find heads specialising in the construction of representations akin to a world model. On the basis of this, we use SAEs (an unsupervised method) alongside supervised classifiers to identify latent features corresponding to the world model. Finally, we use patching experiments and interventions to show that both investigations yield consistent features, and that these form a causal world model with some interesting properties.

## 3.1 WORLD MODEL CONSTRUCTION: CONNECTIVITY ATTENTION HEADS

We began by examining the attention patterns of the maze-solving transformer models and uncovered a notable pattern: in both models, some or all of the attention heads at the first layer ("layer 0") appear to consolidate information about maze connections into the ; context positions. In particular, for all $4i^{\text{th}}$ context-positions tokens (the semicolon separation tokens ; ), these heads attend back 1 or 3 tokens - that is, to one of the two coordinate tokens corresponding to the given connection preceeding the ; token. This pattern is observable for $3/8$ L0 heads in Stan (Figure 3) and $4/4$ L0 heads in Terry (Figure 14). This observation suggests the hypothesis that these heads are in essence constructing a world model for the maze task, for use by later layers.

If this were the case, then we should expect that the output of these heads, mediated by the "OV-Circuit" (Elhage et al., 2021), should consist of combinations of the coordinates captured in a given connection. This can be measured by taking the $W_{OV}$ matrix for each head and measuring the cosine distance between its elements and the model's token embeddings (where coordinate directions are directly given)[1]. With this in mind, we investigated the structure of these vectors more closely. We find an intriguing pattern in the magnitudes of these vectors in the Stan model (Figure 4), while the patterns in Terry were less clear cut (Figure 5).

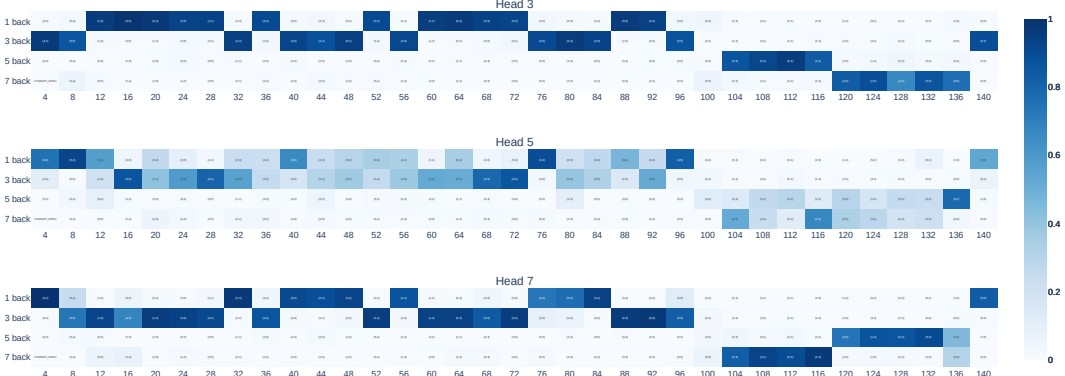

Figure 3: Attention values for heads L0H3, L0H5, and L0H7 in Stan. We use a rather nonstandard representation, looking only at a fixed window into the past of which tokens are attended to by semicolon ; tokens. Every 4th position, up to 140, is shown along the $x$-axis. Color shows attention to positions 1, 3, 5, and 7 earlier in the context (shown along the $y$-axis), for an example 6x6 maze input. This sort of pattern is typical across all inputs examined. Up until context position 100, the heads are attending 1 and 3 positions back; after this the pattern shifts to 5 and 7 back. Note the complementary attention patterns of L0H3 and L0H7. Closer examination shows that L0H3 prefers to direct its attention to 'even-parity' maze cells, with L0H7 preferring 'odd-parity' cells. L0H5 more frequently splits its attention between 1 and 3 back, but sometimes 'fills in' for L0H7. The origins of this pattern are explored further in appendix E; note also the similarities to Figure 4. The other five heads in L0 show no similar pattern. Full patterns are shown in Figure 13

---

[1] As we analyze the first attention layer we can ignore potential "residual drift" in the representations of a given maze coordinate between early and later layers in our transformers (Belrose et al., 2023).

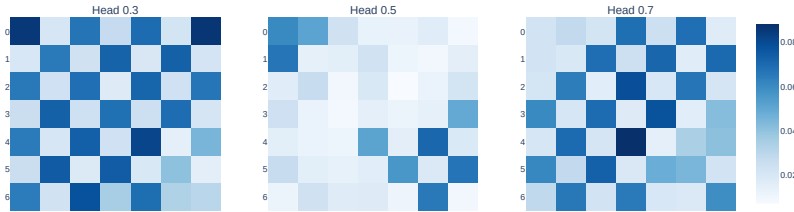

Figure 4: Magnitudes of vectors resulting from applying the $W_{OV}$ matrices of heads L0H3, L0H5 and L0H7 of Stan to maze-cell token embeddings, projected onto the maze grid. The pattern here mirrors the way that the heads divide their attention between the 1-back and 3-back context positions (exemplified in Figure 3) with L0H3 focused on 'even-parity' cells, and L0H7 and LH05 focused primarily on 'odd-parity' cells. This pattern also recurs in the overlaps between query and key vectors of token embeddings, explored in detail in Appendix E.

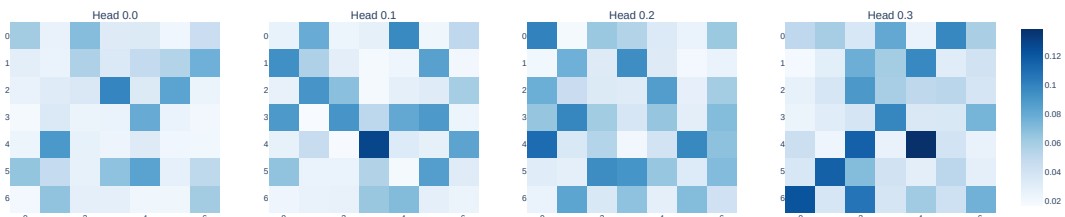

Figure 5: Magnitudes of vectors resulting from applying the $W_{OV}$ matrices of layer-0 heads of Terry to maze-cell token embeddings, projected onto the maze grid. The pattern here is much less striking than that for Stan (shown in Figure 4) although it does suggest that the heads specialise in even/odd-parity cells in localised regions of the maze.

## 3.2 WORLD MODEL REPRESENTATION: SPARSE AUTOENCODERS

As previous work (Ivanitskiy et al., 2024) struggled to intervene on WM features identified via linear probing (Alain & Bengio, 2016), we trained Sparse Autoencoders to attempt to find disentangled features in our models (Cunningham et al., 2023; Bricken et al., 2023). Sparse Autoencoders are motivated by the notion of superposition (Elhage et al., 2022) which posits that artificial neural networks store more features than an orthogonal representation would allow. By training an autoencoder with a higher-dimensional latent space than that of the transformer, tasked with reconstructing a residual stream vector under a sparsity penalty, the hope is that the SAE will recover interpretable features which the transformer was forced to superimpose. Similar approaches have previously seen success on other toy tasks (He et al., 2024; Karvonen et al., 2024).

To prevent "neuron death" in the SAE latent space, resulting from high sparsity penalties, we apply the method of "Ghost Gradients" proposed by Jermyn & Templeton (2024). The resulting trained SAEs faithfully reconstructed the activations (in our case, the residual stream after L0), and replacing these activations with their SAE reconstructed counterparts did not affect model behaviour (Figure 8), giving confidence in the completeness of their representation.

Initial attempts to isolate SAE features corresponding to connections in the maze attempted to use differences in the features present in mazes with or without certain connections. This approach worked well in some cases, but not in others, as not all relevant features varied in magnitude by the same amount, and many features were co-active to a given connection (i.e. those implicated in the path representation, which itself might change when connections are added/removed). To address this, we instead trained decision trees to isolate the relevant features in our transformers (akin to Spies et al. (2022)), as shown in Figure 6.

This analysis yielded our first unexpected finding: Stan's WM consisted of two features for each connection - a somewhat generic "semicolon" feature, as well as a connection specific feature. We visualize highly activating examples of these features in Figure 17, and show that Stan's representation was stable for an additionally trained SAE in Figure 18.

We speculate that this "compositional code" arises in Stan as a result of the transformer imperfectly separating positional information from its WM. This representation also explains why previous efforts to intervene on models with learned positional encodings by using linear probes were unsuccessful - as intervening with a single direction yielded from supervised decoding would also affect the semicolon feature.

It is also interesting to note that Terry encoded connection information very cleanly into single features for each connection - i.e., a single direction in the residual stream. This is in-spite of the fact that Terry's attention heads appeared to operate in a more entangled fashion than those of Stan.

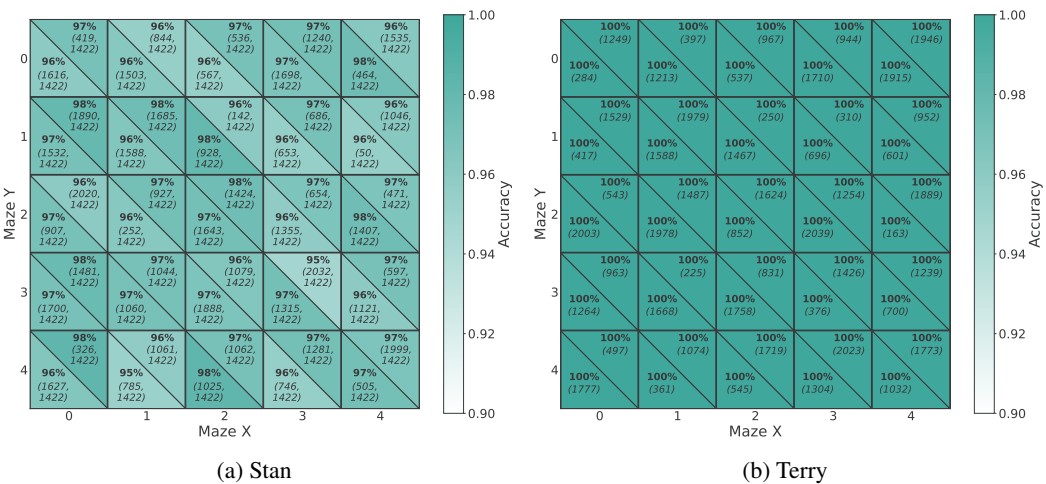

(a) Stan                                           (b) Terry

Figure 6: Decision tree decoding accuracies and relevant features (in parentheses) for each connection in the maze. Upper right triangles correspond to right connections, and lower left triangles correspond to down connections. The decision trees were trained to predict the presence, or absence, of a connection from the SAE feature vector at the semicolon immediately following the definition of that connection. See Figure 17 for more details.

## 3.3 Comparing SAEs and Circuits

In Subsection 3.1 we advanced the claim that certain L0 heads construct features representing maze edges at the ; context positions, specifically by attending to earlier positions containing maze-cell token embeddings, and rewriting those embeddings by application of their $W_{OV}$ matrices. Subsection 3.2 identified features representing maze edges via an independent line of reasoning, by training SAEs, and identifying which of their features were indicative of the presence of a maze edge.

To verify whether these approaches yielded consistent features for the WM, we first calculated the cosine similarity between the features written by isolated attention heads, and those encoded in the SAE (Figure 7a). These showed excellent agreement for Stan, where the attention patterns were clear, but only once the compositional code was taken into account (see Appendix G for details).

Though these results were promising, we carried out a further comparison (Figure 7b) to minimize the assumptions required, and to account for two potential effects: 1) There may be "wiggle room" between feature directions in the model's residual stream, and the circuits that construct them (which would lead to low cosine similarities, even for the same features), 2) As our SAEs are trained after an entire block of computation, it is possible that the MLPs, applied after attention, also played a role in forming the representations. In this second experiment we patched attention head values in the presence of a connection to the mean of a maze set without that connection.By looking at the effect of patching the attention heads on the resulting SAE Latent vectors, we were able to observe that the features considered relevant for any given connection were indeed sensitive to the heads implicated in constructing those features.

In particular, we consider the effect on the SAE features identified in Subsection 3.2 when each attention head is patched at the semicolon position for with its average non-connection value across 500 examples (i.e. removing the contribution a given head toward encoding that connection).

This captures the extent to which a given head contributes towards the "creation" of a maze-connection's representation in the residual stream. These plots not only confirm the link between the attention circuits and the SAE features, but even show the same spatial partitioning of different parts of the maze between different heads. The same plots for Terry, and Stan's down connection, are shown in Appendix F.

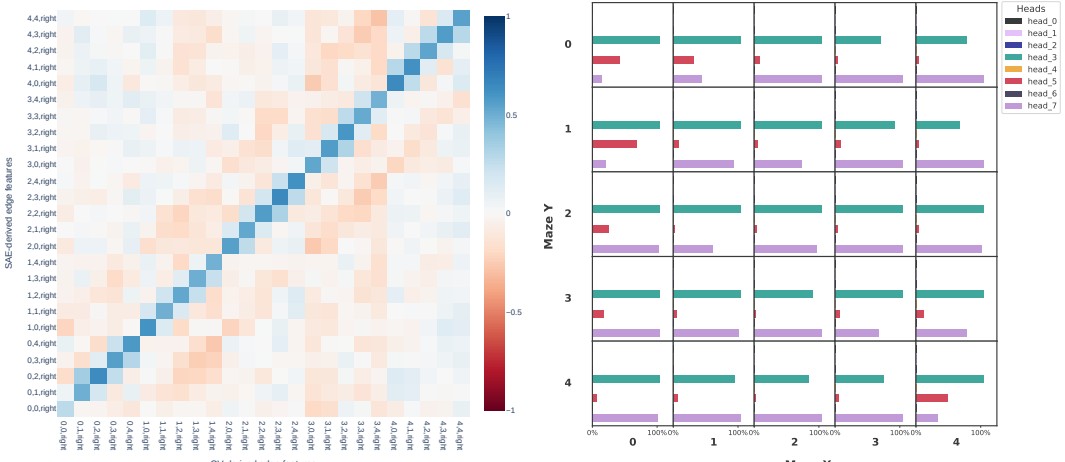

(a) Cosine similarities between edge features derived from the SAE and edge features derived by direct application of the $W_{OV}$ matrices of the heads discussed in Subsection 3.1 to token embeddings for Stan (for reasons of space, only the right directed edge features are shown here, but the pattern is the same for the full set). Details of how features are computed are given in Appendix G.

(b) Effect of patching attention heads on SAE features (rightward connections) for Stan's connection-specific WM features (identified from Figure 6). Compared against Figure 4, we see agreement between the attention analysis and the SAE Feature analysis. We normalize the per-head patching effect magnitudes, such that 100% was the maximal effect seen on an SAE feature's magnitude as a result of patching the attention head (across all features for a given head).

Figure 7: Comparison of SAE features and attention head analysis.

## 4 INTERVENING ON WORLD MODELS

Though a universally agreed upon definition does not exist, we shall consider world Models to be "structure preserving [...] causally efficacious representations" (Millière & Buckner, 2024) of an environment; i.e. representations which preserve the causal structure of the environment as far as is necessitated by the tasks an agent needs to perform. As such, we are interested in understanding how the WMs we have discovered are leveraged by our models to facilitate generation of valid solution sequences. In Figure 8, we give an example of perturbing a feature to "fool" the model into behaving as though it is in a different maze. When patching in the SAE-reconstructed residual stream without perturbations we still see the same behavior as in the original model; when patching in with a modified feature, we see a change in the path. We perform such interventions across 200 examples for each connection feature, and show the resulting intervention efficacies in Figure 9.

The intervention process involves toggling a feature on (to the maximal value observed for that feature in a small dataset) or turning it off (setting it to 0) *at all semicolon positions*[2]. We measure the impact of these interventions on the model's maze-solving accuracy, with a particular focus on how activating versus removing features affects performance. Our results reveal an intriguing asymmetry that constitutes our second finding: interventions that activate features tend to be more effective in altering the model's behavior compared to those that remove features.

---

[2]For the case of adding a connection, this is necessary as there is no semicolon in the sequence which "belongs" to the connection that doesn't exist. We also experimented with toggling to a fixed maximal value in Figure 19, but this was generally less effective. In the case of removal, it made little difference if the feature was disabled everywhere, as it is almost always exactly 0 for a non-matching connection semicolon

This suggests that the transformer may rely more heavily on the presence of certain connectivity cues rather than their absence when constructing its internal world model.

Our final finding relates to the toggling of features in Stan. Though Stan utilized a compositional code, activating the connection-specific features at unrelated semicolons worked in 35% of cases. Conversely, we saw that all removal interventions failed for Stan, for the simple reason that Stan was unable to generalize to sequences containing more connections than it had seen during training - thus failing when shown examples containing the additional connection to be removed (this failure was a result of Stan using learned positional embeddings (Table 1), as shown in Figure 10). The fact that activating connections in the space of the SAE worked at all means that Stan's maze-solving behaviour was at least partially able to generalize in the latent space, where it was decoupled from the positional embeddings.

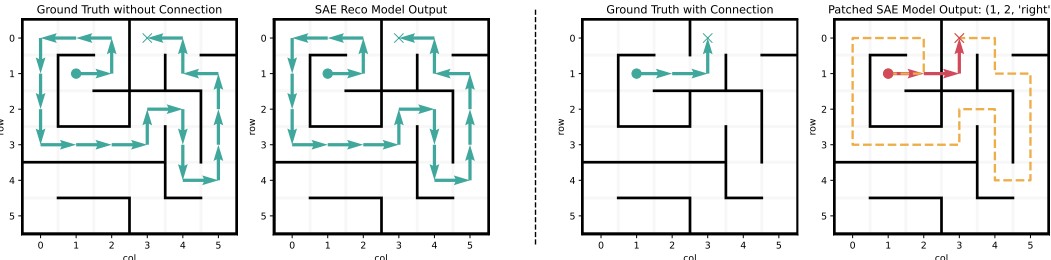

Figure 8: An example of an intervention on Terry where a connection is added by enabling the relevant feature in the SAE's latent space (in this case, feature 250 for `(1,2)` `<-->` `(1,3)` ). From left to right: 1) input maze with ground truth 2) model's prediction with the unperturbed SAE reconstruction patched in 3) perturbed ground truth 4) model's prediction with the perturbed SAE reconstruction in its residual stream at layer 0.

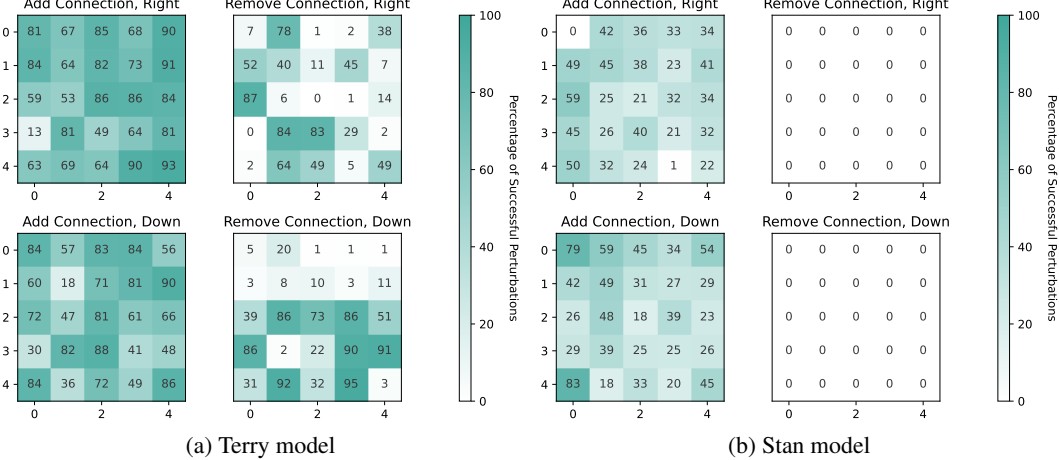

Figure 9: Aggregated accuracy of interventions for examples on which the original prediction was correct. An accurate intervention is one in which the toggling of a connection in the SAE feature space leads the model to act accordingly. Note that Stan removal interventions fail as the inputs in these cases have more connections than the model is able to handle (see length generalization failure in Figure 10).

## 5   RELATED WORK

Our work builds on existing literature in interpretability (Räuker et al., 2023), particularly how transformers develop structured internal representations, often called world models. World Models, as defined by Millière & Buckner (2024), are "structure-preserving, causally efficacious representations of properties of [a model's] input domain."

Here, structure-preserving means that the representations reflect the causal structure of the observation space and causally efficacious means that the model leverages these representations to enable relevant interactions with its environment.

Research into world models has gained traction across various domains, with transformers trained to play complex games like chess being prime examples. For instance, McGrath et al. (2022) trained linear probes to extract various features in AlphaZero's chess model, showing how different aspects of the game, such as piece positioning and potential future moves, are captured within the model's layers. Similarly, Karvonen (2024) investigates the internal representations of a chess model using linear probes and contrastive activations, revealing structured representations of the game state. Jenner et al. (2024) explores the emergence of learned look-ahead capabilities in Leela Chess Zero, where the model encodes an internal representation of future optimal moves.

Another task used to study internal representations in transformers is Othello. Several works have explored the emergence of causal linear world models in this domain Li et al. (2022); Nanda (2023), with recent advancements leveraging SAEs (see Subsection 3.2 to uncover these world models He et al. (2024).

Beyond game-playing tasks, the study of learned world models in transformers extends to other domains, such as natural language processing, where Hewitt & Manning (2019) used probing techniques to uncover the syntactic structure encoded by BERT. This line of research demonstrates that transformer models can implicitly learn hierarchical structures in their residual streams, as explored by Manning et al. (2020). Further supporting this, Pal et al. (2023) demonstrated that the residual stream corresponding to individual input tokens encodes information to predict the correct token several positions ahead, highlighting the model's capacity for structured, anticipatory reasoning.

Additionally, graph traversal as multi-step reasoning has been investigated both from a model capabilities perspective Momennejad et al. (2024) and through mechanistic interpretability Brinkmann et al. (2024); Ivanitskiy et al. (2024), providing further evidence of transformers' ability to encode and utilize structured representations in complex tasks.

## 6 CONCLUSIONS AND FUTURE WORK

In this work, we demonstrated that transformers trained to solve maze navigation tasks form highly structured internal representations that capture the connectivity of the maze and thus act as world models. Through exploratory analysis of attention patterns, we found that connection information was consolidated into semicolon tokens by a subset of attention heads. By using Decision Trees to analyze the latent space of Sparse Autoencoders on these semicolons, we were able to identify sparse features that encoded the position in the maze. We showed that these world models were constructed differently in transformers leveraging learned vs. rotary positional encodings, suggesting that simpler methods such as activation steering or probing would have been insufficient to extract causal world models in at least some cases. More interesting still, we showed that interventions to add connections by toggling features were consistently more effective than interventions that sought to remove connections by zeroing the corresponding features. Furthermore, we found that models with learned position encodings, which were unable to generalize to longer input sequences (i.e., mazes with more connections), were able to behave consistently if additional connection features were enabled via SAE interventions, even if the corresponding token sequence would have caused the model to fail.

These findings shed light on the inner workings of transformers trained on sequential planning tasks and suggest that maze-solving tasks are a rich testbed for understanding the formation of world models in transformers. Future work should aim to uncover whether our findings on intervention asymmetries and steerability are universal - and if not, which conditions give rise to each. An empirical understanding of the reliability of SAE feature discovery and steerability is crucial for AI Safety efforts that attempt to constrain or coerce model behavior through interventions or monitoring based on such methods.

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

# Appendices

## A GENERALIZATION AS A FUNCTION OF INPUT SEQUENCE LENGTH

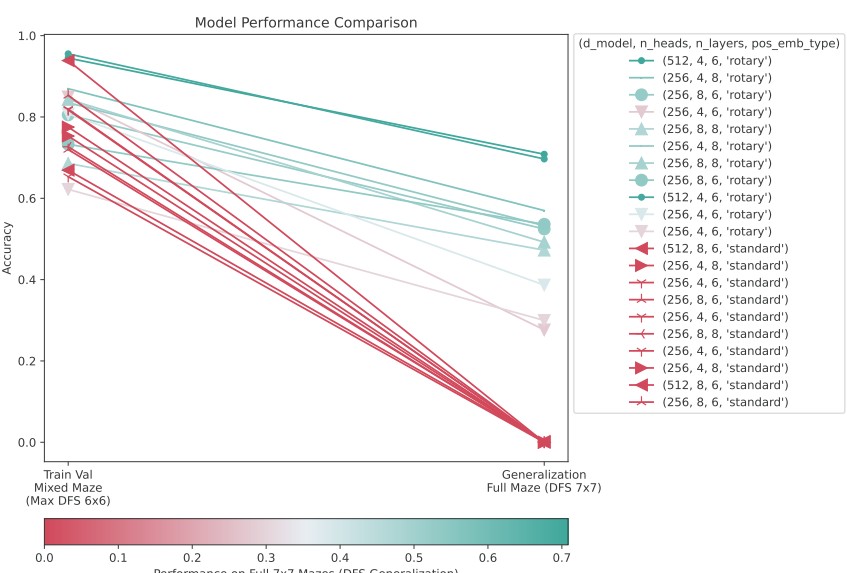

Figure 10: Accuracies of all transformers trained in our sweep on a generalization task. "Train Val" shows the accuracy on the held out in-length-distribution mazes from train time, and "Full Maze" features mazes with more connections (longer input sequences) than those seen at train time. Only rotary models are able to generalize at all

## B SAE TRAINING DETAILS

To choose optimal hyperparameters for our SAEs we ran a sweep over SAEs at layers 2 to 4 on Terry, finding consistent trends across layers. The results of this sweep are shown in Figure 11, and the final details of the SAE analyzed in the main paper are given in Table 2. We also provide feature density histograms for the SAEs analyzed in the main paper in Figure 12 noting that these look good, in that many features are sparse, but also rather distinct from is typically observed in LLMs. This is not surprising, as our token and features distributions will be very distinct from those of natural language, as most mazes have many active connections, and connections are similarly likely to be present in any given maze.

| Sparsity | | | Dataset | | Optimizer | |
|---|---|---|---|---|---|---|
| Expansion Factor | Ghost Threshold | (L0) Sparsity Weight | Batch Size | Training Steps | Learning Rate | Linear Warm Up Steps |
| 4 | 100 | 0.01 | 1024 | $\sim 10^6$ | $10^{-4}$ | 1000 |

Table 2: Hyperparameter values for the final SAEs analyzed in the main paper.

|  | | **SAE Feature Metrics** | | **Average Token Reconstruction Errors** | | |
|---|---|---|---|---|---|---|
| | **Residual Reconstruction Error (L2)** | Sparsity (L1) | L0 | Unperturbed | Zero Patched | SAE Patched |
| Terry | $2.87 \times 10^{-4}$ | 10.7 | 20.9 | 8.18 | 8.52 | 8.18 |
| Stan | $6.35 \times 10^{-4}$ | 9.53 | 28.4 | 6.35 | 8.61 | 6.35 |

Table 3: SAE Metrics for the final SAEs trained on Stan and Terry. We see that replacing the residual stream with the SAE reconstructions has very little impact on the sequence produced by the model, providing confidence that the SAEs are encoding all the relevant information in the model's residual stream.

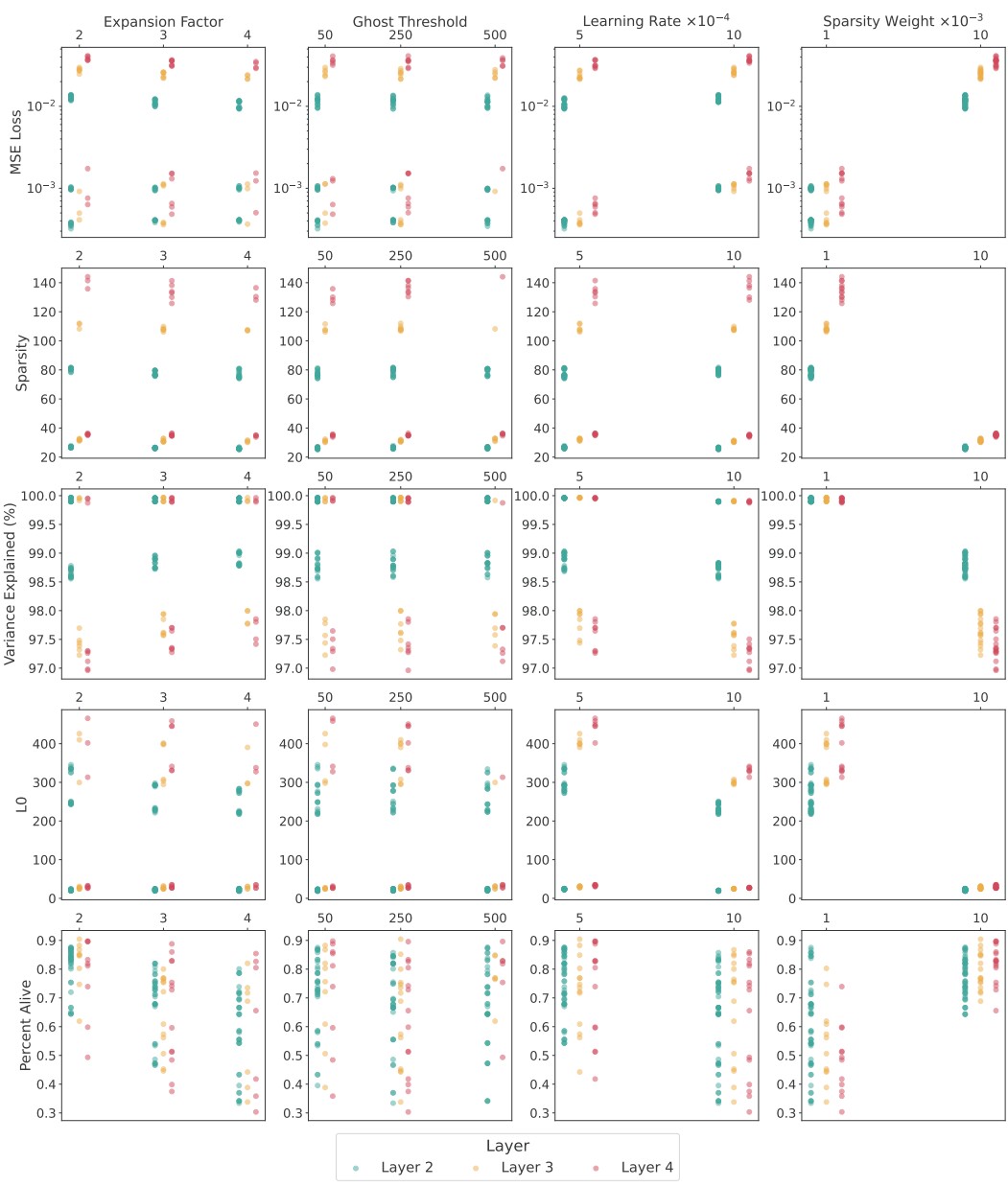

Figure 11: Results of an SAE sweep carried out on Terry.

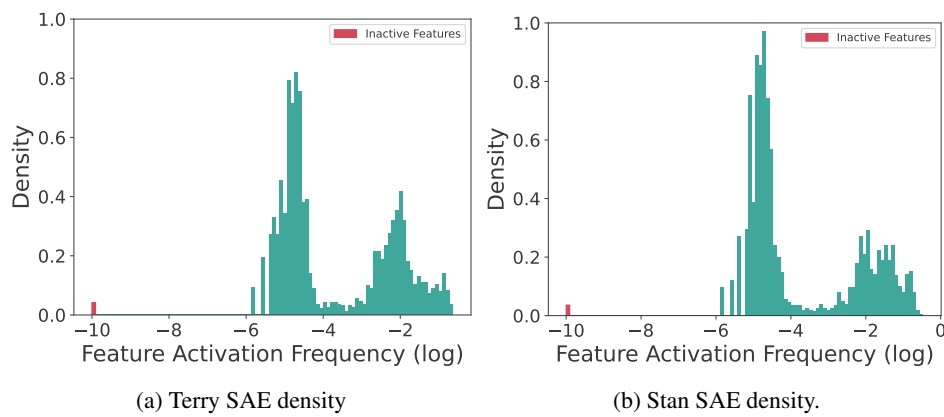

(a) Terry SAE density        (b) Stan SAE density.

Figure 12: Feature density histograms for the SAEs analyzed in the main paper.

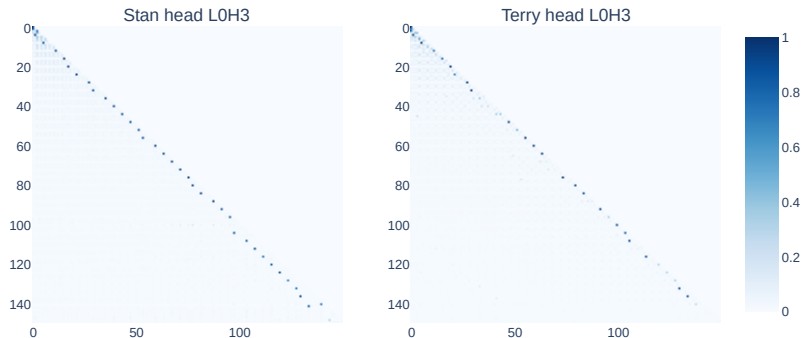

Figure 13: Attention patterns for head L0H3 in Stan and Terry, for a specific example maze. At every fourth context position from 4 through to 140 (the ⟨;⟩ positions in the adjacency-list) attention is directed very strongly back to one or two positions, typically 1 or 3 positions earlier in the context (though for Stan, after context position 100, this shifts to 5 or 7 positions earlier in the context). This pattern is qualitatively repeated across all examples examined, for heads L0H3, L0H5 and L0H7 in Stan, and for all four L0 heads in Terry.

## C    FURTHER ATTENTION VISUALIZATIONS

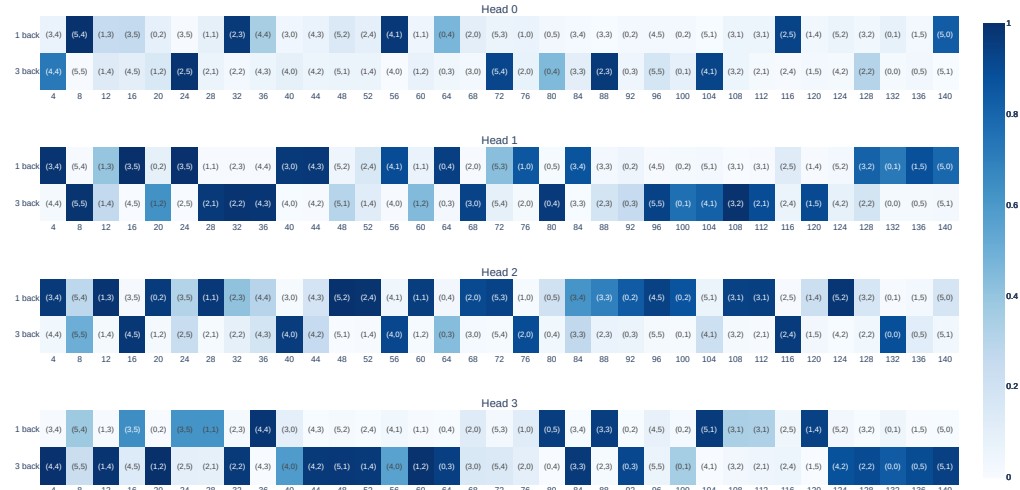

Figure 14: Attention values for layer 0 heads in Terry, from context positions holding the ; token (shown along the $x$-axis) to positions 1 and 3 earlier in the context (shown along the $y$-axis), for an example maze input. This pattern is typical across all inputs examined. The pattern is less clear-cut than for Stan (Figure 3), but note that at every fourth context position, there is at least one head attending strongly to positions 1 and 3 earlier in the context.

# D  HOW SAE REPRESENTATIONS DIFFER

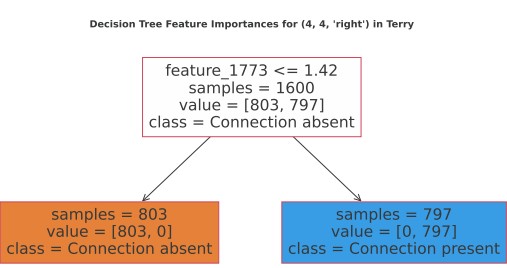
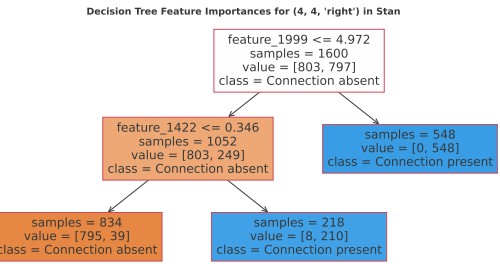

(a) Terry model: A single feature almost perfectly encodes the existence of a specific connection in the maze. This demonstrates the direct encoding of maze connectivity in Terry's SAE latent space.

(b) Stan model: Two features (Feature 1422 and another) work together to encode maze connectivity. Feature 1422 appears consistently across all connections, aligning with the decision tree decoding results presented Figure 6.

Figure 15: Decision trees trained on SAE latents for Terry and Stan models, predicting the existence of specific connections in the maze. These examples illustrate how maze connectivity is encoded in the residual stream at layer 0 on the corresponding semicolon position. The decision trees were trained as supervised classifiers whose target was to predict the presence of a given connection, given an SAE feature vector from the corresponding semicolon position. These SAEs were trained with 10,000 examples per connection (equally balanced between the presence / non-presence of a connection).

**TOP ACTIVATIONS**
**MAX = 4.665**
(5,3) ; (4,4) <--> (4,5) ; (3,0) <--> (3,1) ; (4,1)
(5,1) ; (4,4) <--> (4,5) ; (3,2) <--> (3,3) ; (2,5)
(2,1) ; (4,4) <--> (4,5) ; (5,4) <--> (5,3) ; (2,2)
(0,1) ; (4,4) <--> (4,5) ; (5,2) <--> (5,1) ; (1,2)
(4,4) ; (4,4) <--> (4,5) ; (3,3) <--> (4,3) ; (2,0)

**TOP ACTIVATIONS**
**MAX = 12.012**
(4,2) ; (4,4) <--> (4,5) ; (4,2) <--> (4,3) ; (3,3)
(5,2) ; (4,4) <--> (4,5) ; (1,2) <--> (0,2) ; (3,5)
(1,5) ; (4,5) <--> (4,4) ; (2,4) <--> (3,4) ; (0,5)
(2,4) ; (4,4) <--> (4,5) ; (2,0) <--> (3,0) ; (3,5)
(0,5) ; (4,5) <--> (4,4) ; (1,5) <--> (0,5) ; (4,3)

**TOP ACTIVATIONS**
**MAX = 1.518**
(5,1) ; (3,1) <--> (3,2) ; (2,0) <--> (3,0) ; (3,5)
(5,1) ; (4,5) <--> (3,5) ; (4,1) <--> (4,0) ; (4,3)
(5,1) ; (1,3) <--> (2,3) ; (3,3) <--> (3,4) ; (4,1)
(3,3) ; (5,1) <--> (5,2) ; <ADJLIST_END><ORIGIN_START>
(1,1) ; (5,3) <--> (5,2) ; <ADJLIST_END><ORIGIN_START>

(a) Terry model: A single SAE feature directly encodes a specific maze connection, demonstrating Terry's straightforward representation of maze connectivity.

(b) Stan model: The connection-specific feature activates at the semicolon corresponding to the encoded connection, similar to Terry's encoding strategy (see Figure 17b).

(c) Stan model: Feature 1422, in conjunction with another feature, encodes maze connectivity. This feature appears consistently across all connections, corroborating the decision tree decoding results in Figure 6.

Figure 16: Maximally activating examples, displayed using a modified version of McDougall (2024) for SAE features encoding the connection `(4,4)` `<-->` `(4,5)`, as identified by decision tree decoding. Underlines correspond to loss contribution (blue for positive, red for negative) and highlighting indicates feature activation at a given token position. Connection-specific features in both models (Figure 17b and Figure 17a) show clear activation patterns, while Stan's generic semicolon feature (Figure 16c) exhibits a less obvious trend. Produced using a modified version of McDougall (2024)

## D.1  MAGNITUDE OF INTERVENTIONS

To complement the intervention results presented in the main text, we also conducted fixed-value interventions on both the Stan and Terry models. In these interventions, instead of calculating new activations based on the modified input, we directly set the activations of the targeted features to fixed values. This approach allows us to examine how the models respond to more controlled manipulations of their internal representations.

The fixed-value intervention results shown in Figure 19 reveal interesting patterns that both complement and contrast with the calculated intervention results presented in the main text.

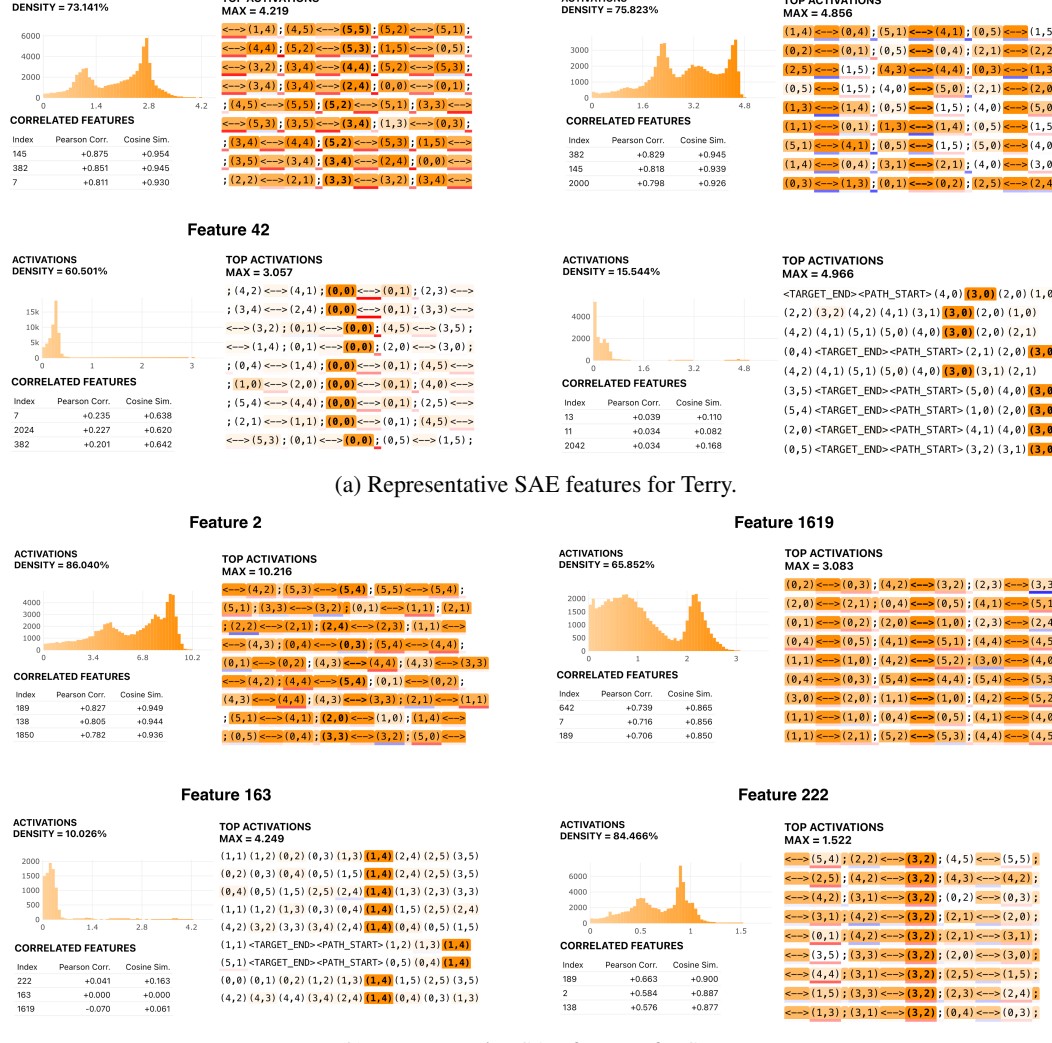

(a) Representative SAE features for Terry.

(b) Representative SAE features for Stan.

Figure 17: We provide examples for the types of features observed in Stan and Terry, beyond the connection features which form the primary focus of the main paper. We observe the same kinds of features between both transformers, and in both cases the predominant features are of the form observed in the top-left (Feature 32 in Terry and 2 in Stan) - These features are more distributed and harder to interpret than the others, and may be suppressed by higher sparsity penalties.

## E  INVESTIGATION OF QK-CIRCUIT IN STAN MODEL

In an effort to better understand the notable "1- and 3-back" attention patterns appearing in heads L0H3, L0H5 and L0H7 of Stan, described in Subsection 3.1, we investigated the query and key vectors for token and positional embeddings, and their overlaps. The scalar products between queries and keys of token embeddings for L0H3 are shown in figure 20. The most striking feature of this plot is the row corresponding to the query vector of the `;` token, and in particular its overlap with the maze cell tokens. Plotting these scalar products on the maze cell grid (figure 21) a clear pattern emerges, analogous to that shown in figure 4, accounting for LH03's tendency to attend to even-parity cells, and LH05's and LH07's tendencies to attend to odd-parity cells. Examining the scalar products among query and key vectors for positional embeddings (figure 22) reveals a pattern that likely accounts for the focusing of attention from `;` context positions to positions 1 and/or 3 earlier in the context.

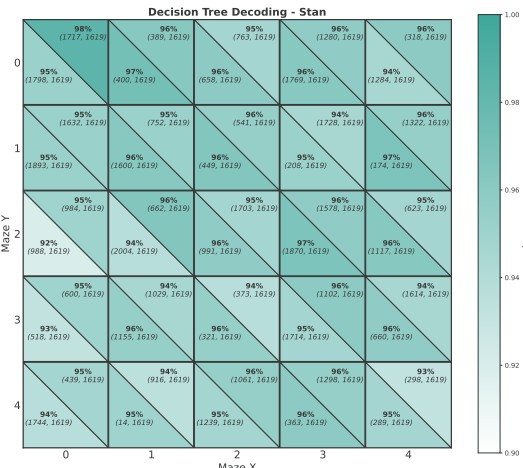

Figure 18: Another SAE trained on Stan gives rise to the same compositional code.

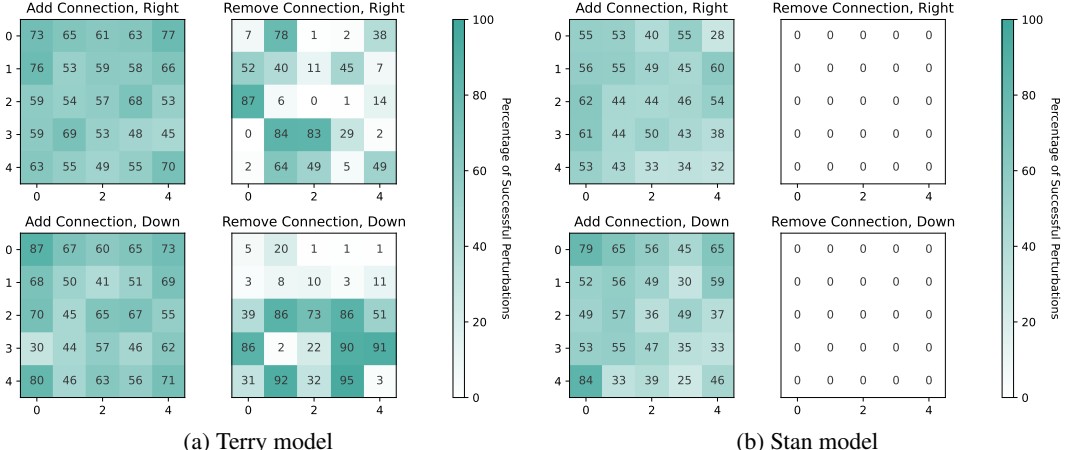

Figure 19: Aggregated accuracy of fixed-value interventions for examples on which the original prediction was correct. As opposed to Figure 9, the addition interventions were performed with a fixed value of 10 (removal interventions were the same, with a fixed value of 0). Here we see that the fixed-value interventions are mostly less effective than the calculated interventions, suggesting magnitude sensitivity for feature magnitudes in the transformer's use of the World Model.

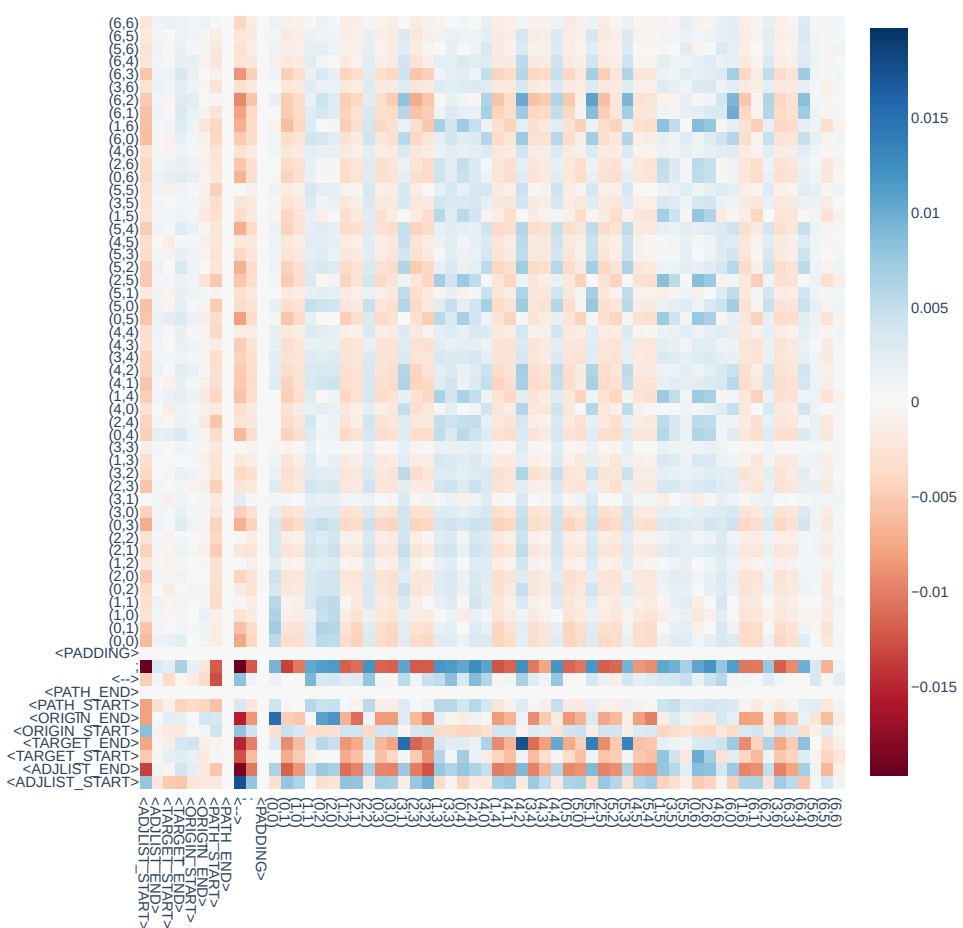

Figure 20: Scalar products of Stan LH03 of query (rows) and key (columns) vectors for token embeddings. Note that the most pronounced pattern is found on the row corresponding to the query vector of the `;` token, reflecting the importance of this head in establishing the attention pattern from context positions containing the `;` token.

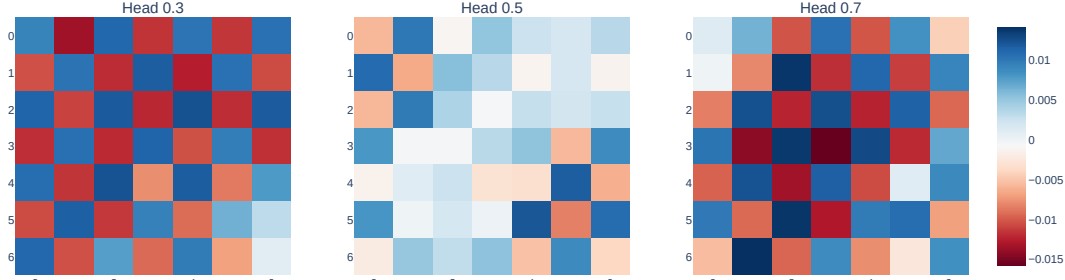

Figure 21: Stan scalar products of query vector for `;` token and key vectors for maze-cell tokens, arranged on maze grid. Note the clear correspondence with Figure 4. These patterns account for why LH03 directs its attention to even-parity cells, while odd-parity cells are attended to by LH07 or LH05.

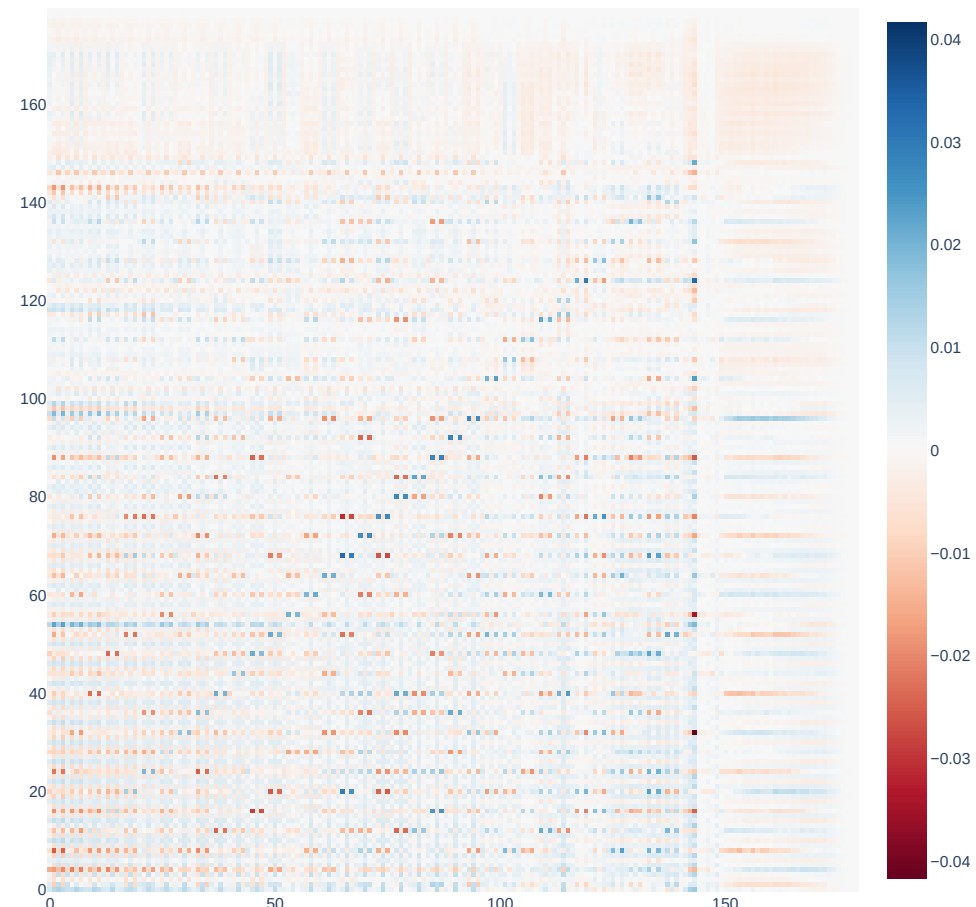

Figure 22: Scalar products of Stan LH03 of query (rows) and key (columns) vectors for position embeddings. Note the approximately diagonal band of pairs of strong positive overlaps every fourth row. This is likely the origin of the '1- or 3-back from `;` ' attention pattern.

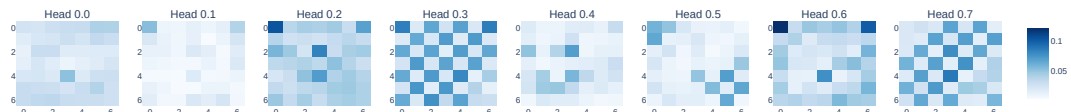

Figure 23: Stan OV projections across position embeddings for all heads.

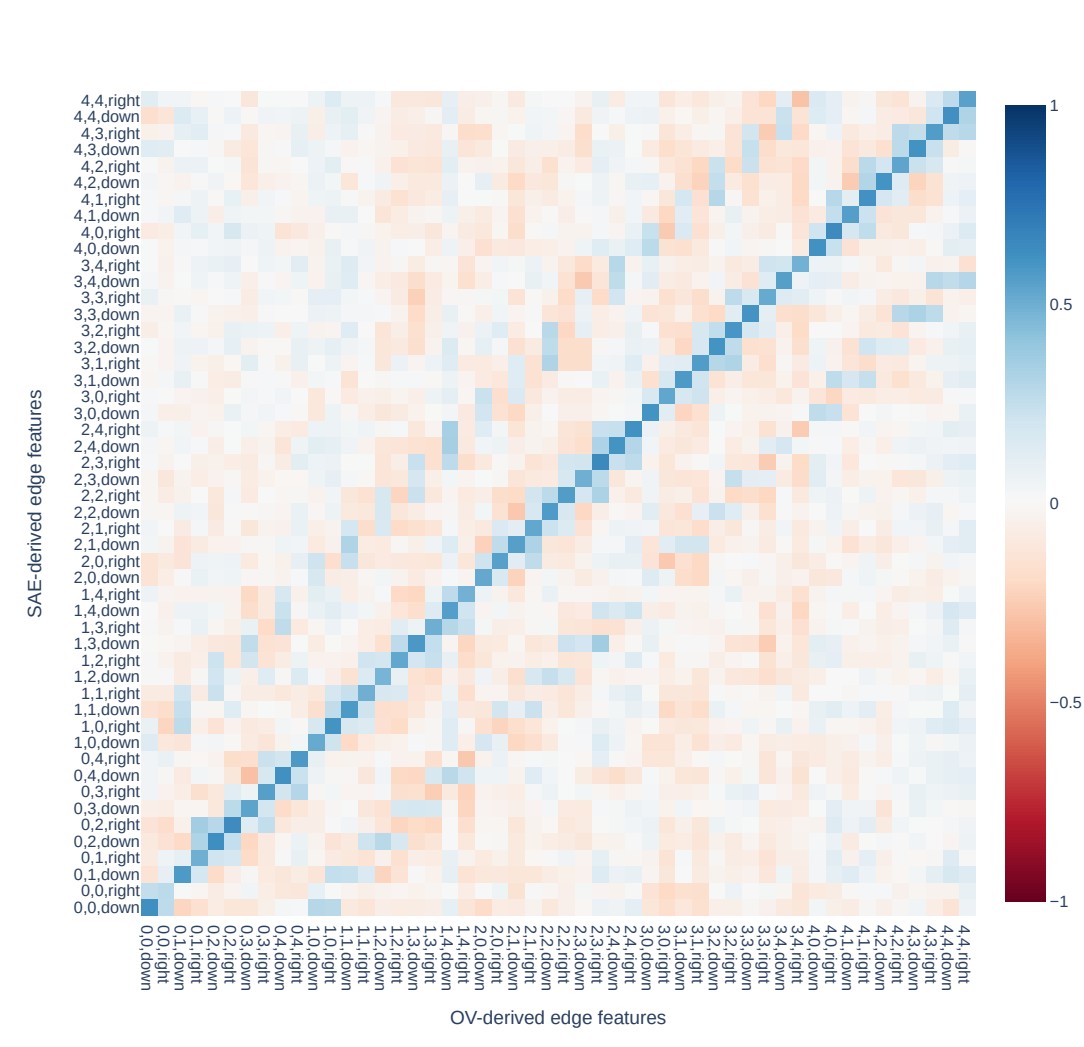

Figure 24: Stan OV-SAE feature similarity for all heads. Complimenting Figure 7a.

# F COMPARING SAEs FEATURES AND CONNECTIVITY ATTENTION HEADS

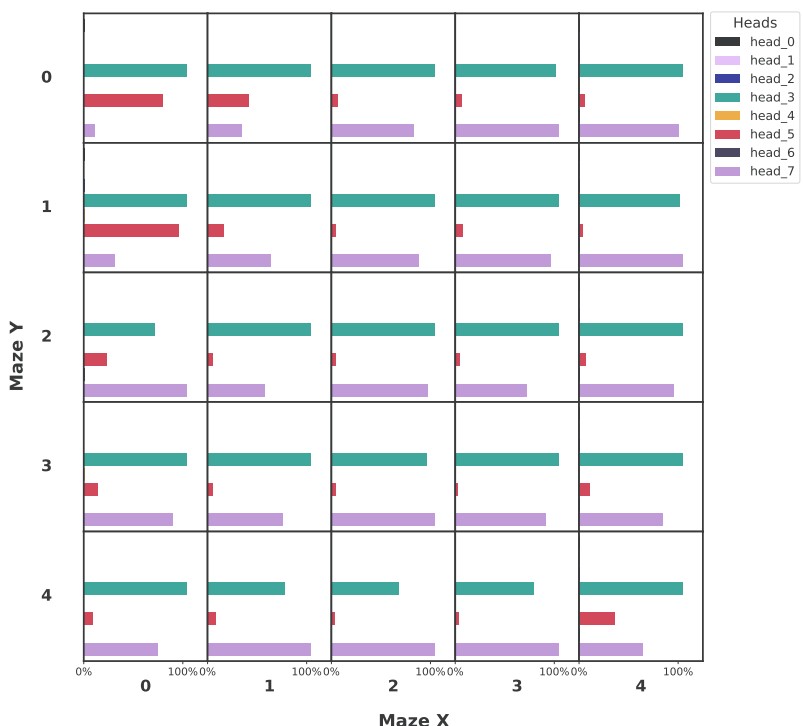

Figure 25: Effect of patching attention heads on SAE features for each down-connection Stan. These again provide agreement with the OV analyses performed in the main text.

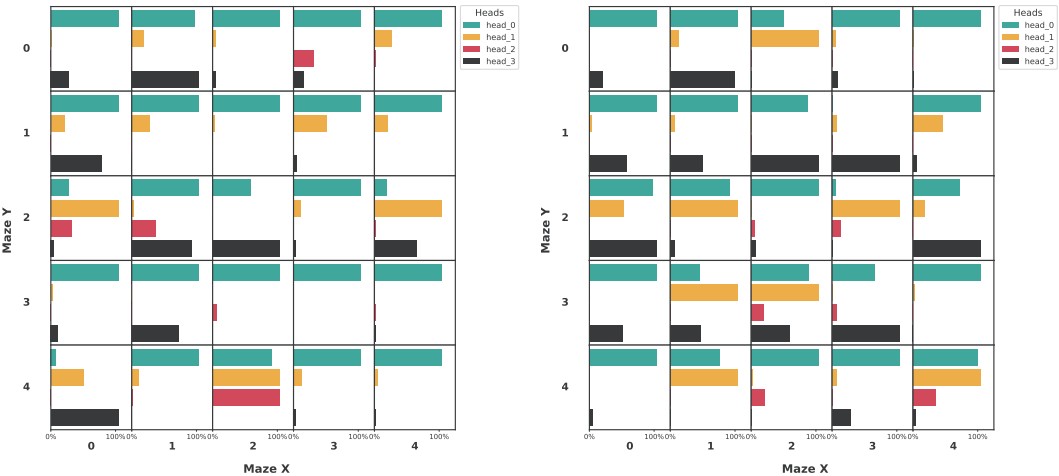

(a) Effect of attention patching on right-connection features.

(b) Effect of attention patching on down-connection features

Figure 26: Effect of patching attention heads on SAE features for Terry. Whilst we observe notable effects, it is difficult to see a clear pattern - as revealed by the attention analyses, the role of each head in constructing a single connection feature in Terry is harder to understand.

## G   COMPUTING SAE AND OV EDGE FEATURE SIMILARITY

In Figure 7a we compute the cosine similarity between SAE edge features and OV circuit edge features.

SAE edge features are formed from a linear combination of the specific edge feature and a "generic edge" feature, with the generic feature coefficient of $-0.6$ being chosen to maximise cosine similarity.

OV edge features are formed from a weighted sum:

$$\sum_{h,c} a_c^h W_{OV}^h t_c$$

Here, $h$ indexes heads L0H3, L0H5 and L0H7, with $W_{OV}^{\mathrm{L0H3}}$, for example, giving the OV matrix of L0H3. $c$ indexes the two cells present in the edge of interest, and $t_c$ is the token embedding of a cell $c$. The coefficients $a_c^h$ are given by the attention directed by head $h$ to cell $c$ from the ; context position following the specification of the edge of interest. Data was averaged averaged over 100 examples (see Figure 3 for one such example).

