# OpenReview forum: "Transformers Use Causal World Models in Maze-Solving Tasks"
_ICLR.cc/2025/Conference — Submitted to ICLR 2025_

### Official Review · Reviewer_1yTd · 2024-10-19

**Soundness:** 1
**Presentation:** 2
**Contribution:** 1
**Rating:** 3
**Confidence:** 4

**Summary:**

The paper attempts to understand the internal representation of a Transformer trained on maze navigation tasks. It mainly studies the internal structure of two models, namely Stan (learned positional encoding + 8 heads) and Terry (RoPE positional encoding + 4 heads).

**Strengths:**

1. Some of the observations (e.g., the attention patterns shown in Fig. 3-4) are interesting.

**Weaknesses:**

1. Overall there does not seem to have a clear picture of the overall internal structured of the learned weights. The paper hits many points but appear disorganized. It is hard to extract the main message.
2. Many claims are not grounded with quantitive evidences, or not conclusive. For example, the paper claims that the Transformer uses "causal world model", but what is the formal definition of the world model in this paper? Does WM mean a collection of correlations shown by attention map? How does such a world model come into play in predicting shortest path? Unfortunately I didn't see clear explanations in the paper.
3. How the decoupling effects of SAE come into play for WM? How is "Isolating irrelevant features" helpful for WM? How to measure the degree of entanglement in the features for Stan and Terry models? The sentences in Sec. 3.2 appear to be very vague.
4. Are any of these observations generalizable to other instance of runs? Fig. 3 talked about specific neurons L0H3, L0H5 and L0H7 and their behaviors such as even-parity and odd-parity. Will such behaviors change over a different runs?
5. What is W_{OV}?

**Questions:**

See above.

---

> ### Author Response · Authors · 2024-12-04
>
> Dear Reviewer 1yTd,
>
> Thank you for taking the time to understand our work and help us to improve its soundness. We have incorporated your suggestions into our work, in particular adding more details in several places and adding a new figure as well as more detail to improve the narrative flow and make our contributions clearer. In addition, we discuss the points you raised below:
>
> 1. **No clear overall understanding of the internal structured of the learned weights:** Existing works in the field do not target a complete understanding of internal weight structure and it is not clear what would constitute a *clear* *overall* understanding. We attempt to thoroughly analyse the features implicated in representing the maze’s connectivity structure within both Terry and Stan through our SAE and attention analyses. Showing a sensible agreement between the circuit (weight level) analysis and the SAE feature analysis is one of our main contributions, and brings us closer to an *overall* understanding of our world models than prior works - though we would not claim that is it yet a *complete* understanding.
> 2. **Regarding the organization of the paper**: We thank the reviewer for highlighting concerns around clarity as a key shortcoming of the initial draft. We have made numerous changes to improve the flow and clarity of our work. Regarding, the contributions - our paper is an empirical investigation into the world models learned by our transformers, which yields a number of interesting findings on the properties of these World Models (along with their existence, which might be considered the implicit central claim). As such, there are multiple contributions, some of which are methodological (efficacy of decision trees and SAEs) and most of which are empirical.
> 3. **“Many claims are not grounded”:**  If the reviewer could list the claims which he finds insufficiently justified we will be happy to respond in more detail. It is worth noting, as we do below, that some of our claims are observations which apply in our context and we do not assert their generality; as such, it is not clear what might be considered “conclusive” evidence beyond what we have shown.
> 4. **“What is the formal definition of the world model in this paper?”:** A mathematical definition of world models is not adopted (and we do not know of any such definition which is considered canonical in the field). However, the paper states what is canonically meant by “world model” and, as such, how the features we discover can be said to constitute a world model. We have attempted to clarify this further in the paper.
>     1.  **“How does such a world model come into play in predicting shortest path”:** This is an excellent question and one we begin to explore in Section 4 - here we show that perturbing the internal representations ****corresponding to the world model causes the transformer to predict paths in-line with having observed a corresponding changed maze in its input. This establishes the causal role of these features and shows that the model uses these internal representations to construct the shortest path. It is, however, beyond the scope of our work to perform a full mechanistic analysis on the question of “how” these perturbations affect representations further downstream (at subsequent layers) in our transformers.
> 5. **How the decoupling effects of SAE come into play for WM?** It is not entirely clear what is meant by “decoupling effects”. We leverage SAEs as a way to recover features of interest from the model’s residual stream without presuming that any of these will correspond to a world model. After training such SAEs with a generic reconstruction objective over the internal residual stream of the model, we then attempt to isolate features relevant to representing the maze (which we dub the “world model”) through the use of decision trees - it need not have been the case that such features were formed by the model.
> 6. **How is "Isolating irrelevant features" helpful for WM?** We assume the reviewer means “isolating **re**levant features”. As the “World Model” refers to a structured internal representation in our transform models, there is no sense in which features which we identify (relevant or not) are “helpful for WM”. The WM either exists, or does not, in the internal representations of our trained models - we merely recover these and attempt to understand their structure and role within the transformers. It is possible that transformers which acquire WMs are better able to perform tasks, but this is not a claim we make in this work.

---

> > ### Author Response · Authors · 2024-12-04
> >
> > 7. **How to measure the degree of entanglement in the features for Stan and Terry models?** This was indeed an important omission - the canonical ways of doing so are through sparsity metrics which we have added to the appendices alongside further SAE training and feature details.
> > 8. **The sentences in Sec. 3.2 appear to be very vague:** We have added thorough training details and further SAE results in the appendices to make our methodology more explicit
> > 9. **Are any of these observations generalizable to other instance of runs?:** We do not have a conclusive answer to this question as analyzing additional sweeps with SAEs requires considerable resources and “feature universality” is not an important part of this paper’s narrative - though we do observe the same sets of features across both Stan and Terry (beyond the Connection features), as shown with new results in the appendices.
> > 10. **What is W_{OV}?** We have added some more detail and pointed to canonical references on the matter of “qk and ov circuits” introduced by Anthropic to the main text. As these notions are standard in the field, we have avoided providing further detail given the existence of excellent references and lack of space in the main text.
> >
> > Thank you again for providing us with your insights on how to improve our work - we hope that in light of the changes we have made to the paper, and the clarifications above, you may see fit to increase your evaluation of our work.
> >
> > The Authors

---

### Official Review · Reviewer_2SjE · 2024-10-30

**Soundness:** 3
**Presentation:** 1
**Contribution:** 2
**Rating:** 5
**Confidence:** 5

**Summary:**

The paper presents a transformer network that learns to solve maze puzzles by constructing the cell connection world model in the first layer. They validate this representation by training an SAE and showing that the SAE cell-connection features are similar to the features written by attention heads in the first layer. The authors also show that the features are causal in adding new connections that are not present in the input. The paper contributes novel insights into how positional encoding can affect the world model representation in transformers.

**Strengths:**

- The claims made in the paper about the maze solving transformer networks are original.
- Most of the claims are well substantiated.
- The analysis of the role of positional encoding in forming the world model is novel.

**Weaknesses:**

Weaknesses:
- The different mechanisms of the Stan and Terry models are not directly comparable as they have different positional encodings as well as different numbers of attention heads. Varying the positional encoding while keeping everything else the same would be helpful in analysing the role of position encoding in the model.
- The result in section 3.1 is qualitative without quantitative support. Adding the average fraction of attention that the ";" token assigns to the previous two cell tokens would provide quantitative evidence.
- Section 4 claims that Stan fails to generalize to mazes with more connections, referencing Figure 10. However, the intervention tests a 6x6 level with one more connection, which should be solvable. Clarifying this discrepancy would improve the argument.
- The paper claims that the model is able to reason with respect to more active features than it would have ever observed during training. However, the experiments only perform interventions that add a single connection feature. The paper doesn't perform interventions that add multiple features simultaneously. Including such experiments would substantiate the original claim.

The writing and the figures are not clear at some places and can be improved. See the clarity improvements provided below.

**Questions:**

Questions:
- Why does the rotary-embedding based model have half the number of attention heads, and hence, half the parameters compared to the standard model? Figure 10 also doesn't have the (512, 8, 6, 'rotary') tuple. Currently, the difference in results between the Terry and Stan models cannot be solely attributed to the change in positional encoding as the number of heads also changes across these models.
- How good is the reconstruction of the SAE? Please provide number like fraction of loss recovered and  explained variance. The paper also doesn't provide the hyperparameters like learning rate, L1 decay, etc. to reproduce the SAE.
- In Figure 4, what does the "magnitude of vector" mean? Do you mean you take the L0 or L1 norm of the vector resulting from applying Wov to a cell-token embedding?
- In section 4, when calculating intervention accuracy on removing a connection, do you check whether the maze is still solvable after removing the connection? If yes, please mention that in the paper. If not, then it is possible that removing the connection, which adds a wall in the maze, makes the level unsolvable.
- Did you find "token-detector" features in the SAE that activate on the maze cell token? If yes, then it is possible that the connection-removing intervention doesn't work because you are not removing the corresponding two maze cell features, which might be used in later layers to detect the presence of their connection.

Improvements for clarity:
- Figure 1(b) is confusing as a visual representation of the maze as it looks like the walls are also maze cells. Please consider replacing it with a visual similar to the one in Figure 8, where the walls are thin and it is easier to identify the cell positions.
- Please increase the font size in all your figures. All the figures have very small x and y-axis labels, which require a high zoom to be legible.
- It is unclear what is being measured in the bar plots of Figure 7b. The text says "effect of patching attention head on the SAE latent vector," but it is not clear how this effect is measured. What does 100% effect mean? Is it measuring the percentage reduction in the SAE latent activation after patching the attention head score?
- The alternating patterns of the L0H3 and L0H7 for Stan (Figure 4) are poorly explained in section 3.1. It only became clear to me after reading section 3.3 and appendix F how the alternating pattern of Figure 4 relates to the flipping attention head pattern between L0H3 and L0H7 of Figure 3.

---

> ### Author Response · Authors · 2024-12-04
>
> Dear Reviewer 2SJe,
>
> Thank you for providing such a thorough review of our work and offering many helpful comments and critiques. We have incorporated as many of these as we were able to into the paper, and provide some further discussion below:
>
> - **Terry and Stan have different numbers of heads:** We chose to analyze terry and stan in spite of this difference simply as they were the most performant models given their position embedding scheme. We understand that this makes it less clear that all differences can be attributed solely to the position embedding scheme, but would make the following two observations which gave us sufficient confidence in analysing these models nonetheless:
>     - The models do in fact have the same number of parameters, up to the difference caused by Stan’s having learnable position embeddings. In our implementation, the dimensionality of the residual stream is fixed, and e.g. decreasing the number of heads means each head has fewer parameters.  (we have added the parameters counts to the paper for completeness, and commented on this fact)
>     - Although Stan has more attention heads than terry, in stan only 3 heads are actually implicated in constructing the world model (connection) features, whilst the others seem to play less clear roles.
> - **Stan’s failure under the addition of a 6x6 intervention:** This was perhaps not sufficiently clearly described - but Stan never saw sequences of length greater than those of a 6x6 fully-connected maze during training and thus fails as soon as even one connection is added (and in Appendix Figure 1 we show also fails for 7x7 fully-connected mazes). It is indeed the case that Stan saw 7x7 sparsely connected mazes at training, but these had adjacency lists shorter than those required by 6x6 fully-connected with an additional connection. As these experiments attempt to remove a connection when the input is a 6x6 maze + 1 connection, Stan failed to generalize.
> - **Addition of more than one feature:** This is a good suggestion, but as we wish to guarantee that interventions affect the shortest path over a balanced and large number of samples, this requires non-trivial dataset generation. We agree that such an experiment with n>1 wall additions would be highly interesting to advance the findings relating to latent interventions which are not possible in the input space, but believe a systematic investigation of the phenomenon of simultaneous intervenability of SAE features, and the asymmetry in addition / removal of features, should be left to future work.
>
> Thank you again for finding our work interesting and providing a thorough, helpful review. We hope that with the above considerations, and the changes made to the paper in mind, you might consider revising your score.
>
> The Authors

---

### Official Review · Reviewer_SDpz · 2024-11-04

**Soundness:** 2
**Presentation:** 2
**Contribution:** 2
**Rating:** 3
**Confidence:** 4

**Summary:**

This paper considers the causal world model emergent from transformers trained on maze-solving tasks. By leveraging attention visualizations and SAE feature attribution, the paper presents that transformers indeed encode the maze structure and sometimes causal depending on the positional-encodings. The paper also proposes the decision-tree method that attributes successfully the features most relevant for edge connections and distinguish the models trained with standard positional embeddings to rotary embeddings.

**Strengths:**

1. Established empirical WM findings in maze-solving transformers.
2. Authors propose the decision-tree method to distinguish features relevant to edge connections, which is novel and useful in this case for insights on the two positional-encoding schemes.
3. There is some cross-checking between the edge features and SAE features in Figure 7a.

**Weaknesses:**

1. Although there was an attempt to do circuit analysis on the transformers, it was limited to attention visualizations and magnitudes of vectors from applying OV matrices. There is not much to take away from this. The authors also speculated on linear probes failure reasons but there is no results on probes. It would be more interesting to corroborate the findings from decision-tree to that of linear probe steering.
2. There is no mention of details of the exact SAE training architecture and procedures as well as a rough sketch of features other than encoding edge connections.
3. Overall, there is not much contribution other than proposing the decision-tree method in this specific setup and the empirical findings on the WMs of maze-solving transformers in this paper do not add too much to that in Igorevich et.al 2023.

**Questions:**

1. In Figure 9, it is mentioned that "Stan removal interventions fail as the inputs in
these cases have more connections than the model is able to handle"; however, if I understand correctly, removing the connections reduce the length so the Stan model should not fail in this case?
2. What is the rough statistics of SAE features and what features other than edge connections are encoded?

---

> ### Author Response · Authors · 2024-12-04
>
> Dear Reviewer SDpz,
>
> Thank you for taking the time to understand our work and to provide us with such valuable suggestions. We’ve made a number of changes to the paper on the basis of these, and provide detailed responses to the points you raised below:
>
> - **Circuit analysis limited to attention visualizations and magnitudes of vectors from applying OV matrices:** Whilst we do observe even/odd parity we were mainly interested in understaning consistency vetween the sae features and the rresults of looking at heads directly. to this end the most interesting part of the circuit analysis was the results of ablating attention heads and seeing which SAE features they affected. The description of these experiments and their corresponding plots were not very clear in the original submission, so we have added detail to make the purpose and findings of these experiments more clear.
> - **No results on probes:** This was an omission as previous work using probes was unable to perform causal interventions, and performing comparable interventions with probes would require knowing exactly what we were looking for; i.e. training probes on latents collected after the first block, only on “;” tokens corresponding to a particular connection, and using other “;”s for negative examples. Whilst is it feasible that this would result in causally effective probes (though less likely so for the transformer’s leveraging compositional codes to encode connections), it is clear that probing here is not an effective way of **discovering** the features that constitute the world model.
> - **SAE Training details and Features**: This was indeed an omission in the first submission. We have now added thorough training details, as well as information about the sweep we performed over SAEs and representative examples of other features discovered in the models.
> - **The paper does not add much to Igorevich et.al 2023.:** Igorevich et al.’s workshop paper demonstrated that the information about mazes was linearly decodable with probing after layer 2 at a single token position. However, there was no demonstration of the causal role of the probes they used. Alongside the use of decision trees, our findings show that transformers in such tasks do acquire a causal world model, but this model is constructed in the first layers and can be understood clearly in terms of subsets of the transformer’s attention heads which specialize in constructing representations of connections.
>
> **Questions:**
> 1. In Figure 9, it is mentioned that "Stan removal interventions fail as the inputs in these cases have more connections than the model is able to handle"; however, if I understand correctly, removing the connections reduce the length so the Stan model should not fail in this case?
> 2. What is the rough statistics of SAE features and what features other than edge connections are encoded: This is an excellent question which we overlooked due to space-constraints. We have added details in the appendices to expand on these findings and show that there are at least four-types of features which SAEs on both models of interest discovered: 1) The connection features which form the WM, 2) specific-token features in the adjacency list, 3) specific-token features in the path (these differ from those in the adjacency list in both models, even though only one model is found to utilize a compositional code), 4) semi-specific features over various coordinates in the adjacency list.
>
> We hope that with these clarifications, and the changes we have made to the paper at your suggestion, you may reconsider your score. We thank you again for helping us to improve the quality of our work.
>
> The Authors

---

### Official Review · Reviewer_gA1f · 2024-11-04

**Soundness:** 2
**Presentation:** 2
**Contribution:** 2
**Rating:** 3
**Confidence:** 4

**Summary:**

The paper discusses the presence of world models in transformers trained on maze tasks. The maze data is given to the model as a series of coordinates and separator tokens including semicolons between maze edges. The authors observe that certain attention heads attend from the semicolon tokens to the coordinate tokens. They also train SAEs on the residual stream and observe that there appear to be features that represent the presence of a single maze edge.

**Strengths:**

The paper is an interesting case study of maze-solving networks, which have not been the subject of much attention in previous interpretability work. The paper furthers the study of world models and SAEs.

**Weaknesses:**

1. My main critique is that it's hard to say that the evidence presented is sufficient to claim that maze-solving networks have a "world model." Previous work (Li et al, https://arxiv.org/abs/2210.13382, Gurnee et al, https://arxiv.org/abs/2310.02207) on world models showed that feeding models data that is derived from some state of the world led models to represent the state of the world itself in a similar way that humans would, in a way that was not necessarily obvious was required for the task. I.e. for humans, we think about world -> data and world -> policy, and training on data -> policy causes the model to represent data -> world -> policy. For example, in Li et al, it's not at all obvious that giving the model data like "E3; G4; etc." for Othello and training it to predict legal moves would cause the model to have a representation of "whether each square is occupied by white or black." In Gurnee et al, it's shown that places are represented with latitudinal and longitudinal coordinates in embedding space, making comparison easy.

In this case, however, it seems obvious that the model should have an atomic representation of whether each edge exists because it's given by the input tokens themselves, and recovering these features does not give us a more interesting representation than the input data itself. In fact, we can use a linear probe to classify the input embeddings directly and still recover the same information (which was not possible in the Li and Gurnee examples, which required the model to execute some nontrivial transformation of the data), so it's unsurprising that this data still exists later in the residual stream.

Similarly, in Section 4, the intervention "create or remove some edge" is comparable to completing an intervention in the model input, so the knowledge of how to edit the world model is not doing something with model internals that we couldn't have done otherwise. (On the other hand, world model edits in the Othello/geography world are not necessarily emulating a straightforward change in the model input and can be more surgical.)

2. I think the fact that the SAEs have a separate feature for each possible edge is a failure mode that occurs when there are too many features. E.g. imagine you have more features than data points, causing every possible activation to have its own feature direction.

3. The writing style and clarity of the paper could be greatly improved. For instance, the use of decision trees needs to be further explained -- is the training objective ablation loss/what features count as the "most relevant"? Additionally, in the present state I do not think Figures 4-5 clearly advance the paper's main arguments.

**Questions:**

1. Could you clarify what the significance is of attention patterns switching from 1/3 preceding tokens to 5/7 preceding tokens after token position 100? I don't think this is sufficiently explained by Appendix D
2. Why should we expect there to be separate attention heads attending to even vs odd parity cells?

---

> ### Author Response · Authors · 2024-12-01
>
> Dear Reviewer gA1f,
>
> Thank you for your thoughtful and detailed review. We appreciate the constructive feedback, particularly regarding the clarity of our writing and methodology. We have made several improvements to address your concerns, which we discuss below.
>
> - Whether the features we find constitute a “world model”: Whilst it is true that the sense in which we claim that connection features constitute a “world model” is not as abstracted from the inputs as those of other related works, this was what motivated our choice of domain - as we wished to form a comprehensive understanding of both how well SAEs are able to capture world-model features and the extent to which identified features agree with those identified from a circuit perspective. We still believe that these features may be meaningfully considered a “world model” in that they form an intermediate representation which is not completely trivial; as we demonstrate, the transformer must learn to combine the tokens which constitute an edge into a single feature - something required to consistently perform well on the task (as our intuitions would have told us) in the absence of naive heuristics or mere path following (which would not require the model to internally represent the connections in full).
> - Whether our interventions are straight-forward / uninteresting: It is indeed the case that our interventions can be reduced to multi-token perturbations, but we do not believe this makes them uninteresting - for two reasons. Firstly, in Othello one could also perturb the input sequence, albeit less trivially, to achieve the desired outcome by using e.g. search over an othello solver. Secondly, even though these interventions may be straight-forward there are still interesting behaviours which we observe. I.e.:
>     - There is an asymmetry in adding / removing features (which would not be the case with input perturbations).
>     - It is possible to e.g. add features (toggle additional connections on) to the Stan model and coherently effect its maze solving behaviour in the presence of these. However, the model fails to generalize to longer sequences, so attempting the same intervention in the input space, by e.g. adding four tokens specifying the edge, would lead the model to fail 100% of the time (as we demonstrate).
>
>     As such, whilst the content of the interventions is by the choice of environment straightforward, we observe behaviours which we consider non-obvious.
>
> - Our SAEs have too many features / the features are not atomic: This is a valid concern, especially given the lack of detail we provided on our SAE training and evaluation in the initial submission. We have added our validation results and representative examples of SAE features to the appendices, and make the following observations:
>     - The features occur after one transformer block on the semicolon position, and are not mere semicolon-demarkers as they have a causal role in model behaviour and capture the maze structure.
>     - The SAEs intentionally form an overcomplete basis but through sparsity penalties are decentivised from encoding redundant information. Furthermore, metrics measuring sparsity and feature density distributions give reasonable values.
>     - We observe other SAE features which display clear structure, as we have shown with the added figures in the appendices.
> - We have also improved the writing in places to hopefully improve the clarity of the work. We have also provided further details on the use of decision trees in the appendices to answer your question regarding the target - they are trained in a fashion similar to the linear probes of prior work, with a supervised classification target for the presence or absence of an edge (thus isolating the edge features).
>
> **Questions:**
>
> - On the significance of patterns switching from 1/3 back to 5/7 back: You are right to point out that this is not well-explained, and indeed we have no strong explanation as to why our transformers decided to varying circuits, besides high-level arguments around symmetry breaking and the likely fact that these may be learned at different times during training. Pending additional experiments, we leave further investigation to future work as this does not significantly concern the focus of our paper.
> - Regarding even/odd parity attention heads: This is a very interesting question which we do not have a good answer to! This was a finding which we did not particularly expect, but which is not too surprising after-the-fact. At a high-level we believe this related to subspaces of differing attention-heads at the same layer being maximally orthogonal to reduce interference, but leave a complete investigation to future work.
>
> We hope that in light of the changes to the paper, and the clarifications we have attempted to provide in the discussion above, you might reconsider your scores. Once again, we are grateful for your help in improving our work.
>
> The Authors

---

### Meta-Review · Area_Chair_2zrg · 2024-12-20

**Metareview:**

This paper focusses on 'World Models' (WMs) in transformers trained on maze tasks. They use SAEs to show that features are similar to those in the first layer attention heads features. They also intervene causally and find different effects.

Reviewers agreed that this paper is not ready for publication. Unfortunately, many reviewers did not engage with the author rebuttal, and so kept their 'reject' recommendation. I think that the paper is stronger after rebuttal, but still not ready for acceptance. Another issue is that the authors seem to have made many changes during the rebuttal period (eg to address clarity issues brought up by reviewers), which is hard to verify in a rebuttal period, and is better judged as a new submission to a different venue.

Reviewers agreed that maze-solving tasks is an interesting place to study these effects.

A common theme is that it the evidence in the paper is not enough to say there is a world model (Reviewer gA1f, and 2SjE after rebuttal / during discussion), using the definition of WM from other works. The author rebuttal was not enough for me (I still view this as a limitation of the paper); perhaps this should be actively brought up early in the paper to avoid confusing a reader. I thought the authors rebutted gA1f's point on straightforward interventions, and it is good to see they added more details about SAEs. I also agree with Reviewer 2SjE on providing detailed explanations for the differences between Stand and Terry networks, as the current explanation is not enough. Reviewer 1yTd seems to have difficulties understanding the paper narrative; the authors say they have updated the paper to fix many of these issues, and I have not checked this all, so am paying much less attention to this reviewer's score (1yTd did not engage in discussion after the rebuttal).

**Additional Comments On Reviewer Discussion:**

The authors rebutted many points, especially to do with clarity and writing. Please see metareview for more details. The only reviewer that engaged with author rebuttal is Reviewer 2SjE, who still thinks that the paper is (marginally) below the acceptance threshold, despite the paper being clearer and easier to follow now.

---

### Decision · Program_Chairs · 2025-01-22

Reject